# A Building Extraction Approach Based on the Fusion of LiDAR Point Cloud and Elevation Map Texture Features

**Xudong Lai** [1,2] ⓘ**, Jingru Yang** [1] ⓘ**, Yongxu Li** [1,*] ⓘ **and Mingwei Wang** [2,3] ⓘ

[1]  School of Remote Sensing and Information Engineering, Wuhan University, Wuhan 430079, China
[2]  Key Laboratory for National Geographic Census and Monitoring, National Administration of Surveying, Mapping and Geoinformation, Wuhan 430079, China
[3]  Institute of Geological Survey, China University of Geosciences, Wuhan 430074, China
*  Correspondence: Liyongxu@whu.edu.cn; Tel.: +86-131-6462-5398

**Abstract:** Building extraction is an important way to obtain information in urban planning, land management, and other fields. As remote sensing has various advantages such as large coverage and real-time capability, it becomes an essential approach for building extraction. Among various remote sensing technologies, the capability of providing 3D features makes the LiDAR point cloud become a crucial means for building extraction. However, the LiDAR point cloud has difficulty distinguishing objects with similar heights, in which case texture features are able to extract different objects in a 2D image. In this paper, a building extraction method based on the fusion of point cloud and texture features is proposed, and the texture features are extracted by using an elevation map that expresses the height of each point. The experimental results show that the proposed method obtains better extraction results than that of other texture feature extraction methods and ENVI software in all experimental areas, and the extraction accuracy is always higher than 87%, which is satisfactory for some practical work.

**Keywords:** LiDAR point cloud; building extraction; elevation map; Gabor filter; feature fusion

## 1. Introduction

Remote sensing is the acquisition of information about objects or phenomena without physical contact [1]. A large amount of remote sensing data has been generated and applied, and improvements in the spatial and temporal resolution of remote sensing images have made them become the main data source for object extraction [2], such as tree crown extraction [3], coastal zone detection [4], road recognition [5], etc. Buildings constitute the main component of urban areas, and building extraction using remote sensing images has become a hot research topic as remote sensing technology has the advantage of being fast, large-scale, and economical. Some researchers provided information of the spectral, geometrical, contextual, and rooftop segment patch via the morphological building index (MBI) and saliency cue to extract building information, which had good performance and versatility under different image conditions. However, the image-based building extraction technique is limited by large intra-class differences and small inter-class differences in spectral features [6,7]. 3D information is valid for buildings, especially elevation information, while for the image. it is complicated to realize, and it is mainly reflected by the change of elevation, which is important information of buildings.

As one of the active remote sensing data sources, LiDAR uses laser pulses to measure the distance between the sensor and different objects. It is widely used in geodesy [8], geo-statistics [9], archeology [10], geography [11], the control and navigation of autonomous vehicles [12], etc. Compared with 2D images, which only provide position and shape information, LiDAR can conveniently acquire

3D information on objects in terrain. Therefore, many studies have applied the LiDAR point cloud to conduct building extraction [13]. Wang et al. adopted a building extraction technique based on the point voxel group by using the class-oriented fusion method and "horizontal hollow ratio", which was effective for large-scale and complex urban environments [14]. Qin et al. demonstrated the use of geometric and radiation features of the waveform and the point cloud with parametric and non-parametric classification methods. The experimental results suggested that it was efficiently used for urban land cover mapping [15]. Zhao et al. utilized connected operators to extract building regions from LiDAR data, neither producing new contours, nor changing positions, which was effective, and the average offset values of simple and complex building boundaries were 0.2–0.4 m [16]. Huang et al. proposed a novel object and region-based top-down strategy to extract buildings, and the experimental result proved that the proposed method achieved good performance and was robust when parameters were within reasonable ranges [17]. Yi et al. detailed a method for reconstructing the volume structure of urban buildings directly from the original LiDAR point cloud. The experimental results demonstrated the advantage of the approach in terms of effectiveness on large-scale and raw LiDAR point data [18].

However, the discreteness of the point cloud may lead to the loss of some features, and it is difficult to distinguish objects with similar heights, while it is able to extract different objects with texture features in 2D images. As elevation map is a kind of 2D image obtained by projecting the point cloud onto 2D planes, and it can provide abundant texture features and has been utilized in the field of building extraction. Fasahat et al. realized building extraction by transforming the point cloud into an elevation map and analyzing gradient information from the elevation map. Experimental results showed the effectiveness in eliminating trees, extracting buildings of all sizes, and extracting buildings with and without a transparent roof [19]. Liu et al. combined remote sensing data of multiple sources to draw height maps of different object types for land cover and land use mapping, which coincided well with the ground survey data with an accuracy of 5.7 m by root mean squared error (RMSE) [20]. Kang et al. achieved the rendering of barren terrain by enhancing the geometric features of elevation maps and increased the number of landscape features, which was most suitable for rendering barren terrain or planet surfaces [21]. He et al. proposed to organize LiDAR point data as three different maps: dense depth map, height map, and surface normal map. It was proven to recover successfully object hierarchies, boundary sharpness, and global integrity regardless of point cloud sparsity, large loss, and 3D to 2D degradation uncertainty [22]. In addition, the texture feature extraction method can be used to obtain features to extract objects on the basis of the elevation map, which can robustly detect buildings from satellite images and outperforms state-of-the-art building detection method [23]. Cao et al. constructed a unified multilevel channel characteristic framework and realized target detection based on histograms of oriented gradient (HoG) features. The experimental results showed that this method could reduce the missed detection rate and improve the detection speed [24]. Du et al. used the gray level co-occurrence matrix (GLCM) features to obtain textures from an elevation map and combined them with point cloud information to achieve area and object-level building extraction, and the results suggested a good potential for large-sized LiDAR data [25]. Niemi et al. inventoried soil damage from forwarding trails and fitted a logistic regression model for predicting the event of soil damage, which showed that DTM-derived local binary patterns (LBP) were useful in terrain trafficability mapping [26].

Point cloud information can reflect the spatial structure of ground objects, but its discrete type may lead to the lack of correlation information of each part. Texture features can reflect the correlation of each part and help to distinguish different objects. Therefore, point cloud and texture features can be fused to achieve complementarity, as well as reflect the features of objects from multiple dimension so as to obtain better results. However, the increased data dimension may lead to an increase in time complexity, and feature selection is always utilized to solve the problem. As the essence of feature selection is a combinatorial optimization problem, which means selecting a satisfactory feature subset to conduct building extraction, it is usually solved by swarm intelligence algorithms [27]. In this paper,

by fusing the point cloud and texture features, as well as conducting feature selection, a building extraction technique is realized. Point cloud features are extracted based on the eigenvalue, density, and elevation, and the point cloud is also transformed into an elevation map to extract texture features. After that, the fusion of the point cloud and texture features is used to extract buildings from different experimental areas. Among various swarm intelligence algorithms, particle swarm optimization (PSO) is easy to implement,and stably converges to the optimal solution. Therefore, it is adopted to obtain the superior feature subset for building extraction in the paper.

This paper is structured as follows. In Section 2, the core method and basic principles of this paper are elaborated in detail. The steps of the method are described in Section 3. Section 4 describes the experiments that are carried out according to the method, the experimental data, the final results, and the evaluation of the accuracy. Section 5 summarizes the work of this paper and research prospects.

## 2. Basic Theory of Gabor Filters

As for 2D images, the Gabor filter is one of the efficient filtering techniques and is based on a sinusoidal plane wave. Its use has been explored in many applications [28,29]. The Gabor filter can not only characterize the spatial frequency structure of an image, but also retain spatial relationship information, and the spatial frequency positioning ability is essential to extract orientation-dependent frequency content from the pattern [30]. Furthermore, as the Gabor filter is invariant to zoom, rotation, and translation, it is suitable for texture representation and recognition [31]. In the spatial domain, a 2D Gabor filter is a Gaussian kernel function modulated by a sinusoidal plane wave, which consists of a real part and an imaginary part representing the orthogonal direction. These two parts can either form a plurality or be used separately [32,33].

The formula for the Gabor filter is expressed as below:

$$g(x,y) = \left(\frac{1}{2\pi\sigma_x\sigma_y}\right) exp\left(-\frac{1}{2}\left(\frac{\bar{x}^2 + \bar{y}^2}{\sigma_x^2 + \sigma_y^2}\right) + 2\pi jW\bar{x}\right) \tag{1}$$

$$\bar{x} = xcos\theta + ysin\theta \quad \bar{y} = -xsin\theta + ycos\theta \tag{2}$$

where $\sigma_x$ and $\sigma_y$ are parameters that describe the spread of the current pixel in the neighborhood in which weighted summation occurs, $W$ is the central frequency of the complex sinusoid, $\theta \in [0, \pi)$ is the orientation of the horizontal to vertical stripes in the equation above, and æ represents the imaginary unit.

The extraction of texture features using the Gabor filter includes two main processes: filter design and the effective extraction of texture feature sets from the filter's output. The process of acquiring texture features from an image using the Gabor filter is as follows. Firstly, the input image is divided into blocks. Secondly, the Gabor filter banks are established, and thirdly, we convolve the Gabor filter templates with each image block in the spatial domain; each image block obtains the filter outputs. These outputs are of the image blocks' size. Fourthly, each image block is passed through the outputs of the Gabor filter templates and is "condensed" into the texture feature of the image block [34].

## 3. Building Extraction Based on the Fusion of Point Cloud and Texture Features

### 3.1. Point Cloud Features

At first, as the LiDAR system generates a number of noise points when acquiring data, which is usually manifested as elevation anomaly points and will affect the accuracy of building extraction, the point cloud is denoised, and elevation anomalies are filtered out. After that, the features of the point cloud, which include various eigenvalues, are obtained. Unlike eigenvectors, eigenvalues have good rotationally-invariant properties [35], and therefore, feature extraction based on the point cloud's eigenvalues was used for building extraction. Besides, density and elevation are both critical attributes of point cloud. Thus, features based on eigenvalues, density, and elevation were extracted as the

reference data of building extraction. The specific meanings and formulas used in the calculations are shown in Table 1.

**Table 1.** Point cloud features.

| Category | Name | Abbreviation | Meaning | Formula |
|---|---|---|---|---|
| Eigenvalue-based features | Sum | SU | Sum of eigenvalues | $\lambda_1 + \lambda_2 + \lambda_3$ |
| | Total variance | TV | Total variance | $(\lambda_1 \lambda_2 \lambda_3)^{1/3}$ |
| | Eigen entropy | EI | Characteristic entropy | $-\sum\limits_{i=-3}^{3} \lambda_i \cdot In(\lambda_i)$ |
| | Anisotropy | AN | Anisotropy | $(\lambda_1 - \lambda_3)/\lambda_1$ |
| | Planarity | PL | Planarity | $(\lambda_2 - \lambda_3)/\lambda_1$ |
| | Linearity | LI | Linearity | $(\lambda_1 - \lambda_2)/\lambda_1$ |
| | Surface roughness | SR | Surface roughness | $\lambda_3/(\lambda_1 + \lambda_2 + \lambda_3)$ |
| | Sphericity | SP | Sphericity | $\lambda_3/\lambda_1$ |
| Density-based feature | Point Density | PD | Point Density | $0.75 * \frac{N_{3D}}{\pi r^3}$ |
| Elevation-based features | Height above | HA | The height difference between the current point and the lowest point | $Z - Z_{min}$ |
| | Height below | HB | The height difference between the highest point and the current point | $Z_{max} - Z$ |
| | Sphere Variance | SPV | Standard deviation of the height difference in the spherical neighborhood | $-\sqrt{\frac{\sum\limits_{i=1}^{n}(Z_i - Z_{ave})^2}{n-1}}$ |

$\lambda_1$, $\lambda_2$, and $\lambda_3$ are eigenvalues of the point cloud, where $\lambda_1 < \lambda_2 < \lambda_3$. An analysis of the eigenvalues and eigenvectors can often provide important information for extraction decisions. According to the points in the neighborhood, the covariance matrix of the center point was calculated, and then the eigenvalues of the point were obtained. Based on these eigenvalues, 12 kinds of features can be calculated, including sum of eigenvalues (SU), total variance (TV), eigenvalues (EI), anisotropy (AN), planarity (PL), linearity (LI), surface roughness (SR), and sphericity (SP). AN refers to the uniformity of the point distribution on three arbitrary vertical axes, which helps to separate anisotropic structures, such as power lines and buildings, from vegetation. PL is a measurement of planar characteristics of the point cloud, and planar structures have high PL values. As the surface of a building's roof reflects laser directly, this feature is remarkable. LI is a measurement of the linear attributes of a point cloud. The power lines and edges of buildings have obvious linear structures, and the linearity of these points is characterized by high values. SR is the average number of points allocated by the point cloud in three directions. The distribution of vegetation points in all directions has no tendency, so the SR values of vegetation are high. The density of the point cloud in the neighborhood of penetrable targets, such as vegetation, reflects the distribution of the point cloud and is usually higher than that of buildings. In the vicinity of the cylinder at the center point, the height differences, including that between the current point and the lowest point (height above (HA)) and that between the highest point and the current point (height below (HB)), were calculated. The standard deviation of the elevation included the elevation of each point in the spherical and cylindrical neighborhoods. $Z_{ave}$ is the average of the current neighborhood's interior point elevation; $n$ is the number of points in the current neighborhood's interior point cloud; and $Z_i$ is the $i$-th point in the neighborhood. The sphere variance (SPV) value is high for objects with few changes in elevation [36]. For high-rise building facades and roofs, the differences between the current and the lowest elevation are usually much larger than that of other points. Therefore, building facades can be distinguished effectively, while the standard deviations of elevations in spherical neighborhoods can be used to identify ground and other horizontal planes. All of the mentioned features above show the properties of point clouds from the point of view of eigenvalues, elevations, and densities. They can provide more effective information for building extraction than single-scale features [37].

### 3.2. Texture Feature Extraction Based on the Elevation Map

In this study, the point cloud was transformed into an elevation map for texture feature extraction. The process of transformation was based on the elevation distribution of the point cloud, and it was

easy to operate. Firstly, the grid size was set as 1 m, and then, the point cloud was rasterized according to its x and y coordinates, while each grid corresponded to one pixel in the elevation map [38]. After that, the height threshold was set, and the elevation variance of all points in the corresponding grid of each pixel was calculated. If the variance was below the threshold, the average elevation of points in the gird was selected as the gray reference value of the corresponding pixel. Otherwise, the height distribution curve was interpolated based on triangulation in the natural neighborhood, and half of the peak value was taken as the gray reference value. For a grid with few or even no points, the median elevation value of the points in the K-nearest-neighbor was taken. After all the above, the gray reference values were normalized to 0–255. In this way, the elevation map corresponding to the point cloud can be obtained as shown in Figure 1.

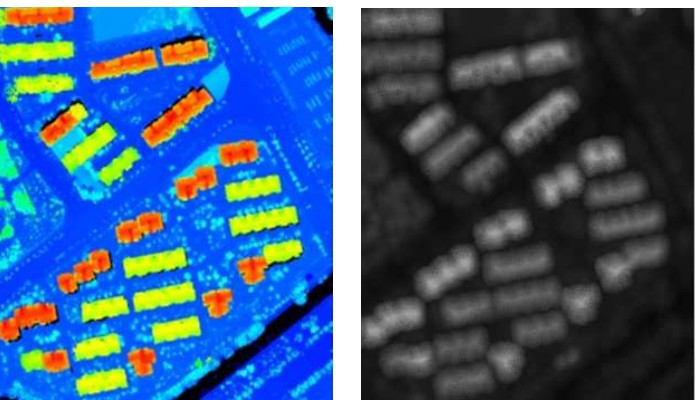

**Figure 1.** Point cloud rendering image (left) and elevation map (right).

After the elevation map was obtained, the corresponding texture features were extracted for further building extraction. Compared with other methods, the Gabor filter can capture those features that correspond to different spatial frequencies (scales) and orientations, so it can be used to discriminate features of images. In this study, a 2D Gabor filter was used to extract texture features. The texture features of elevation maps in different orientations and scales were obtained by changing the values of the orientation and frequency parameters. The orientation and frequency values were updated as follows:

$$\theta(i) = \frac{(i-1)\pi}{O}, \ where \ i = 1, 2, ..., O \tag{3}$$

$$f(i) = \frac{f_{max}}{\sqrt{(2)^{i-1}}}, \ where \ i = 1, 2, ..., S \tag{4}$$

where $\theta(i)$ is the orientation parameter, $O$ is the number of orientation parameters, $f(i)$ is the frequency variable, and $S$ is the number of frequency variables. In this study, four frequency values and six orientation values were combined to obtain 24 texture features. The frequency values changed gradually with 0.2, 1.414, 0.1, and 0.0707. The Gabor filter convolution kernel functions were in six different orientations: 0, $\pi/6$, $\pi/3$, $\pi/2$, $2\pi/3$, and $5\pi/6$ with the same frequency value.

### 3.3. Feature Selection for Reducing the Number of Features

In this paper, PSO was used for feature selection to decrease the data dimension, which is a kind of swarm intelligence algorithm using a group of particles. It has been noted that members of a group seem to share information among themselves, which is a fact that leads to increased efficiency of the group. A particle moves toward the optimum based on its present velocity, its previous experience, and the experience of its neighbors. In an n-Dimensional search space, the position and velocity of the $i$-th particle are represented as vectors $X_i = x_{i1}, ..., x_{in}$ and $V_i = v_{i1}, ..., v_{in}$. Let *Pbest$_i$* and *Gbest* be

the best position of the *i*-th particle and the group's best position so far, respectively. The velocity and position of each particle are updated as follows [39]:

$$V_i^{k+1} = \omega \cdot V_i^k + r_1 \cdot c_1 \cdot (Pbest_i^k - X_i^k) + r_2 \cdot c_2 \cdot (Gbest^k - X_i^k) \tag{5}$$

$$X_i^{k+1} = X_i^k + V_i^{k+1} \tag{6}$$

where $V_i^k$ is the velocity of the *i*-th particle at iteration $k$, $\omega$ is the inertia weight factor, $c_1$ and $c_2$ are the acceleration coefficients, $r_1$ and $r_2$ are random numbers between zero and one, and $X_i^k$ is the position of the *i*-th particle at iteration $k$. In the velocity updating process, the values of the parameters such as $\omega$, $c_1$, and $c_2$ should be determined in advance, which makes it cumbersome to solve large-scale optimization problems.

However, decimal coding may not be suitable for discrete optimization such as feature selection; thus, the position vector of a particle should be coded as a binary form. The velocity of the *i*-th element in the *i*-th particle is related to the possibility that the position of the particle takes a value of one or zero. It is implemented by defining an intermediate variable $S(v_{ij}^{k+1})$, called a sigmoid limiting transformation, as follows [40,41]:

$$S(v_{ij}^{k+1}) = \frac{1}{1 + exp(-v_{ij}^{k+1})} \tag{7}$$

The value of $S(v_{ij}^{k+1})$ can be interpreted as a probability threshold. If a random number selected from a uniform distribution in [0,1] is less than the threshold, the value of the position of the *j*-th element in the *i*-th particle at iteration $k + 1$ (i.e., $x_{ij}^{k+1}$) is set to one, and otherwise to zero, and the position vector is replaced as follows:

$$x_{ij}^{k+1} = \begin{cases} 1 & if \quad rand < S(v_{ij}^{k+1}) \\ 0 & otherwise \end{cases} \tag{8}$$

where *rand* denotes random numbers uniformly distributed between zero and one; $S(v_{ij}^{k+1})$ is a sigmoid limiting transformation.

In this paper, PSO was used to extract as high an accuracy as possible with few features. To improve the training process, the feature combination was adjusted by PSO, and the optimized results could be obtained by choosing the feature combination with the minimum error as the most suitable one [42]. Finally, a reasonable combination of point cloud and texture features was obtained for building extraction, and the whole process is shown in Figure 2.

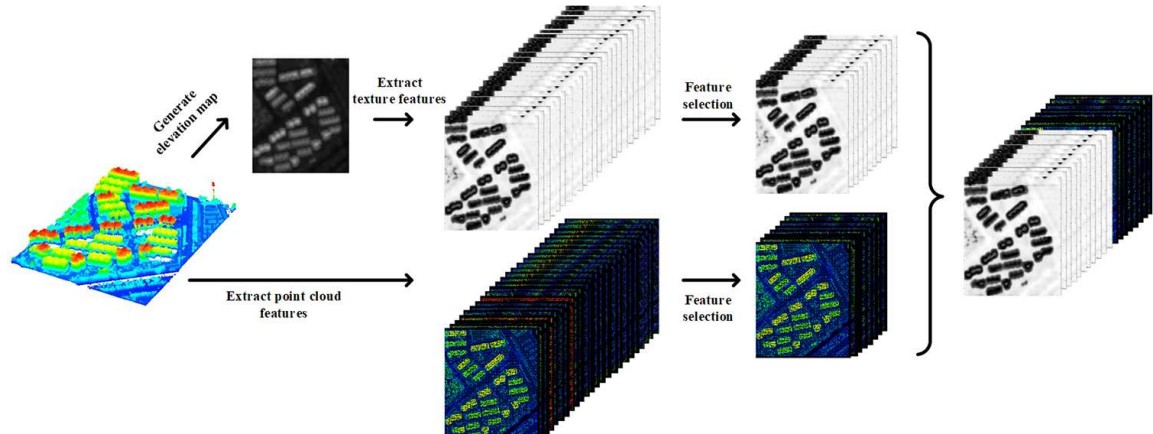

**Figure 2.** The process to obtain the optimal combination of features.

### 3.4. Definition of the Objective Function

To obtain the results of high extraction accuracy and reduce the number of features, an objective function was defined as an auxiliary in this paper. As the Fisher discriminant criterion has been shown to have good performance in building extraction and other extraction problems that include two categories, maximizing the differences between classes and minimizing the differences within classes, and accurately identifying the target category from other classes, it was used to define the objective function for feature selection [43]. The formula of the objective function is expressed as follows:

$$fit = \frac{(\mu_1 - \mu_2)^2}{(\sigma_1^2 + \sigma_2^2) \cdot n} \tag{9}$$

where $fit$ represents the value of the objective function, $\mu_1$ and $\mu_2$ are the eigen mean vectors of two types of objects, $\sigma_1$ and $\sigma_2$ are the eigen variance vectors of two types of objects, respectively, and $n$ is the number of points. The output of point cloud features was the vectors. The texture features were in the form of a 2D image. They can be converted into vectors, and finally, these two kinds of features can be merged into a vector, while a higher feature vector dimension of each point can be obtained by combining the two vectors. Besides, the larger value of the objective function demonstrated better quality of classification.

### 3.5. Implementation of the Proposed Method

The proposed method was easy to implement, and the key issues of building extraction were the fusion of point cloud and texture features, as well as feature selection. The process of the proposed method is shown as follows:

- Step 1: Input the testing images, and compute the feature vectors of the point cloud. Generate elevation maps, and extract texture features via the Gabor filter from them.
- Step 2: Build the training and testing samples based on the fusion of point cloud and texture features;
- Step 3: Randomly generate the initial population of PSO in the range of $-10$–10 via decimal coding, and transform it into binary coding;
- Step 4: Conduct building extraction, and compute the fitness value of each particle by Equation (9);
- Step 5: Operation of PSO:

  Step 5-1: Update the velocity of each particle by using Equation (5);

  Step 5-2: Switch the population into the form of binary coding by Equation (8);

- Step 6: Conduct building extraction, and compute the fitness value of each particle by Equation (9);
- Step 7: If the solution is better, replace the current particle; otherwise, the particle does not change, and then, find the current global best solution;
- Step 8: Judge whether the maximum number of iterations is reached, and if it is, go to Step 9; otherwise, go to Step 5;
- Step 9: Output the optimal feature combination, and compare it with other building extraction methods via the extraction accuracy.

## 4. Experimental Results and Discussion

The experimental environment in this study was a computer with a 2.30-GHz CPU and 8 G of RAM. The data-processing operation was realized using MATLAB 2016a and VS2017 software. The manual extraction process was accomplished using LiDAR software and visual interpretation by researchers with relevant working experience.

*4.1. Experimental Platform and Data Information*

The data used in this study were point cloud data obtained from a Riegl LMS-Q780 laser scanner in Fuzhou, China. The experimental data included five non-overlapping urban areas, which contained buildings, vegetation, and other types of objects. Since the high density of the experimental point cloud may result in a large amount of calculation, it was necessary to down-sample the data in order to reduce the amount of calculation. According to the density of the point cloud after down-sampling, the data areas were divided into Low-Density Region 1 (LDR 1), LDR 2, the medium-density region (MDR), High-Density Region 1 (HDR 1), and HDR 2. Details on the experimental data are shown in Table 2.

**Table 2.** Experimental data information. LDR, low-density region; MDR, medium-density region; HDR, high-density region.

| Experimental Data | Data Area (m²) | Number of Points | | Point Cloud Density | |
|---|---|---|---|---|---|
| | | Original Data | After Dilution | Original Data | After Dilution |
| LDR 1 | 174,080 | 4,486,763 | 19,320 | 25.799339 | 0.111040 |
| LDR 2 | 155,595 | 3,989,310 | 21,926 | 25.683631 | 0.140958 |
| MDR | 186,147 | 585,024 | 23,675 | 26.261592 | 0.183575 |
| HDR 1 | 99,470 | 2,283,275 | 29,127 | 23.062170 | 0.294197 |
| HDR 2 | 68,040 | 1,897,760 | 20,663 | 27.936171 | 0.303810 |

The experimental data were colored according to the elevation rendering, and the results of the manual extraction are shown in Figures 3–7.

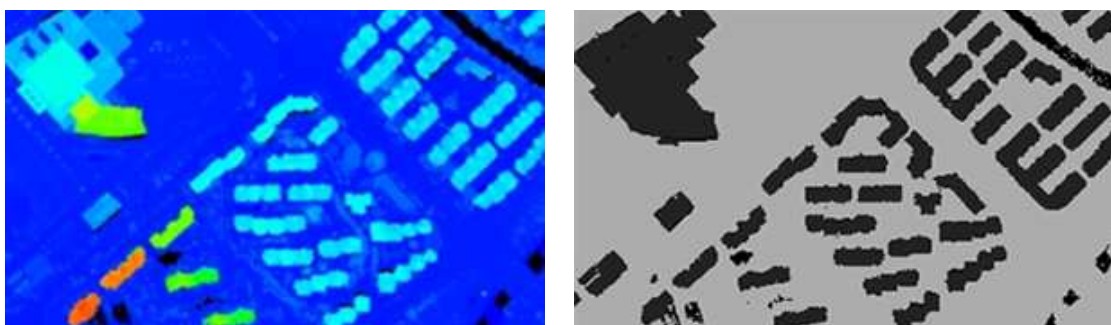

**Figure 3.** LDR 1 elevation coloration and manual extraction results.

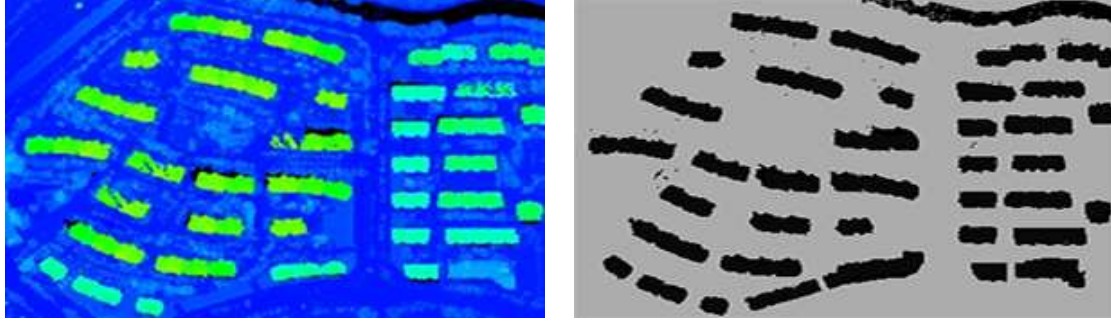

**Figure 4.** LDR 2 elevation coloration and manual extraction results.

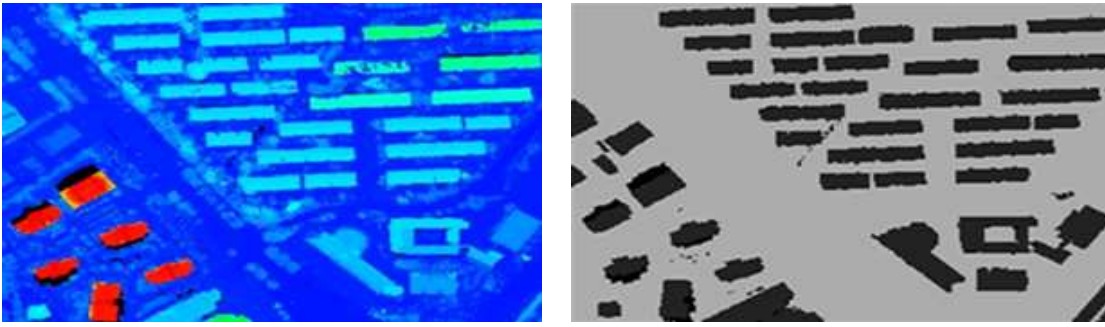

**Figure 5.** MDR elevation coloration and manual extraction results.

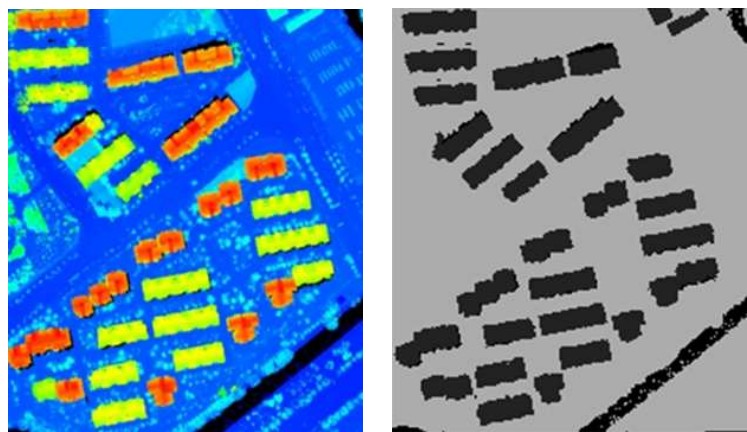

**Figure 6.** HDR 1 elevation coloration and manual extraction results.

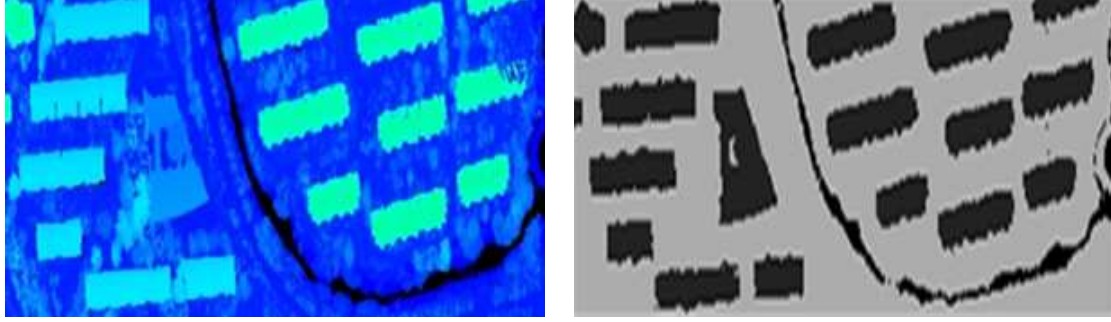

**Figure 7.** HDR 2 elevation coloration and manual extraction results.

*4.2. Extraction of Texture Features*

The process of extracting texture features using the Gabor filter in this study is shown below:

Figure 8 shows the process of the Gabor filter, where it is formed on the basis of different values of the orientation and frequency parameters. Different texture features can be yielded after elevation map convoluting with templates. A group of parameter combination results is shown in Figure 8 with the same frequency value of 0.2, and the orientation varied from 0–$5\pi/6$ via steps of $\pi/6$, while the local display of the common part is also shown on the right side. It can be concluded that variation of the parameter combination caused the change of the convolution module and resulted in differences in texture features, especially on the edge and corner of the buildings.

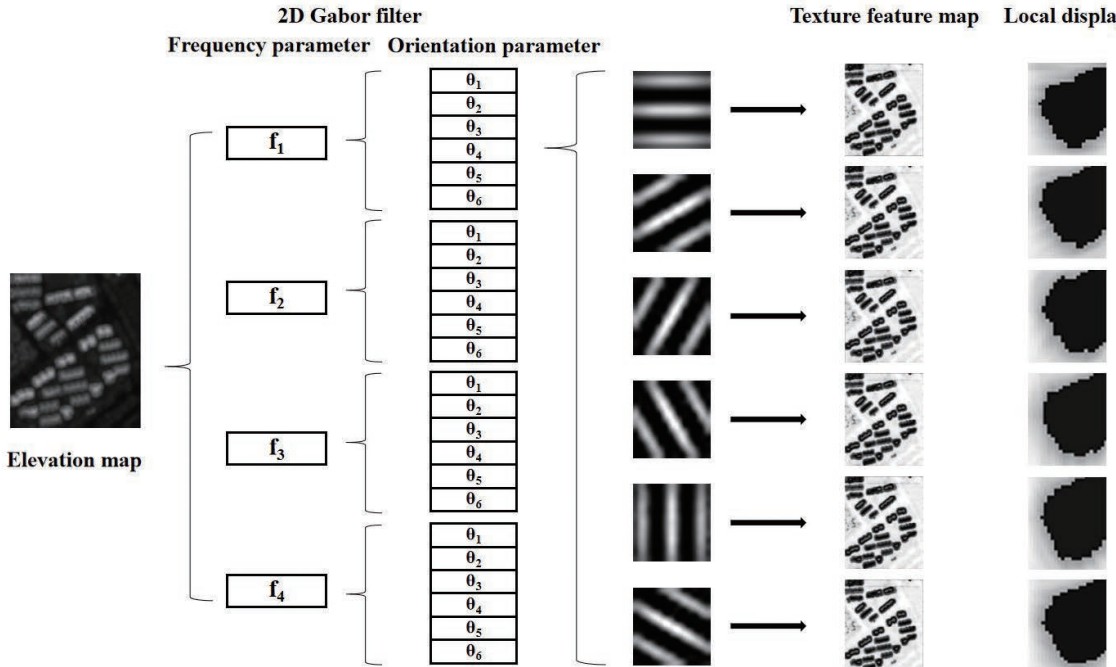

**Figure 8.** Texture features extraction using the Gabor filter.

### 4.3. Comparative Analysis and Accuracy Evaluation of Building Extraction

In order to prove the effectiveness of the proposed method, the experimental results were compared with those obtained using GLCM, LBP, and HoG for texture feature extraction. Those of building extraction based only on point cloud features (OPCF), building extraction with no feature selection (NFS), and building extraction using ENVI software were also compared in this paper. The extraction accuracy of different methods is shown in Table 3, and the building extraction locations of different experimental areas are shown in Figures 9–13.

**Table 3.** Comparison of the experimental results with other methods for building extraction (%). OPCF, only point cloud features; NFS, no feature selection.

| Experimental Data | GLCM | HoG | LBP | OPCF | NFS | ENVI | Proposed |
|---|---|---|---|---|---|---|---|
| LDR 1 | 86.9984 | 75.9503 | 88.3870 | 80.4586 | 78.7330 | 87.4203 | 90.4238 |
| LDR 2 | 65.5523 | 85.1865 | 74.5297 | 85.5651 | 89.6949 | 91.3310 | 92.2558 |
| MDR | 75.8902 | 78.9356 | 73.3347 | 81.7022 | 82.3527 | 83.5180 | 87.1679 |
| HDR 1 | 87.5064 | 90.8264 | 90.0470 | 87.4961 | 81.6047 | 90.2660 | 92.1138 |
| HDR 2 | 62.3917 | 76.6975 | 75.2795 | 79.4367 | 84.2762 | 86.2752 | 89.1207 |

From Table 3, it can be seen that the extraction accuracy with the proposed method was superior to other texture feature extraction methods and ENVI software. For HDR 2, the extraction accuracy of the proposed method was over 10% higher than that of GLCM HoG and LBP. Especially for LDR 2 and HDR 2, the extraction accuracy of GLCM was only around 60%, while the proposed method could still achieve an extraction accuracy higher than 87%. Although the extraction accuracy by using ENVI software exceeded 80%, and even reached 90% for LDR 2 and HDR 1, the extraction accuracy of the proposed method could still be 1.2874% higher than ENVI software. Comparing with the result of NFS, this suggested that feature selection benefited the building extraction by improving its extraction accuracy and efficiency. In all, when the follow-up operations were the same, the final results obtained by the Gabor filter applied in this paper were more accurate than those of other texture feature extraction methods. After feature selection, not only the extraction accuracy was higher, but

also the computational time was shorter, as the data dimension was decreased. Besides, using about 10 features, we were able to achieve such satisfactory results.

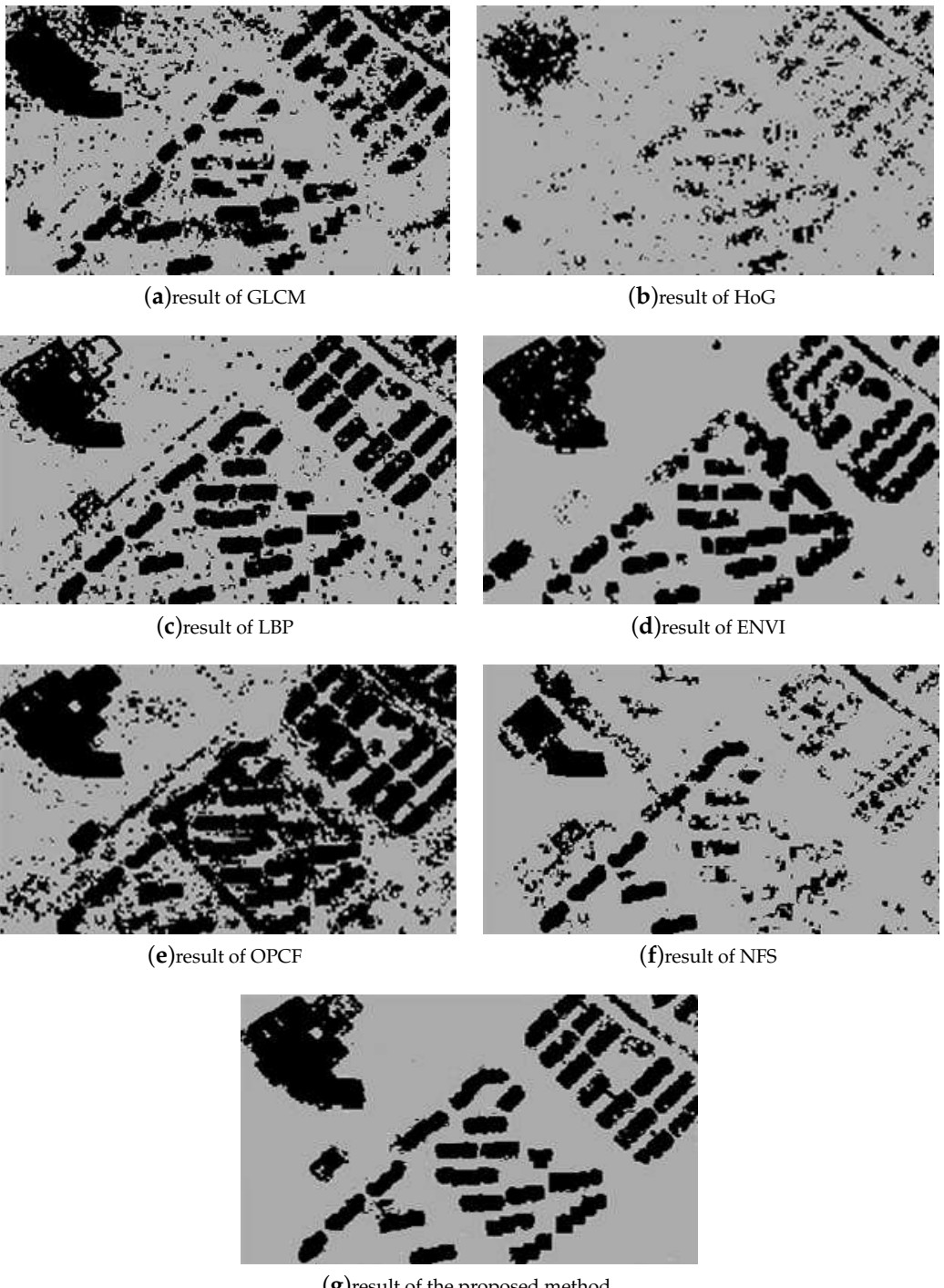

(**a**)result of GLCM

(**b**)result of HoG

(**c**)result of LBP

(**d**)result of ENVI

(**e**)result of OPCF

(**f**)result of NFS

(**g**)result of the proposed method

**Figure 9.** Building extraction results of LDR 1.

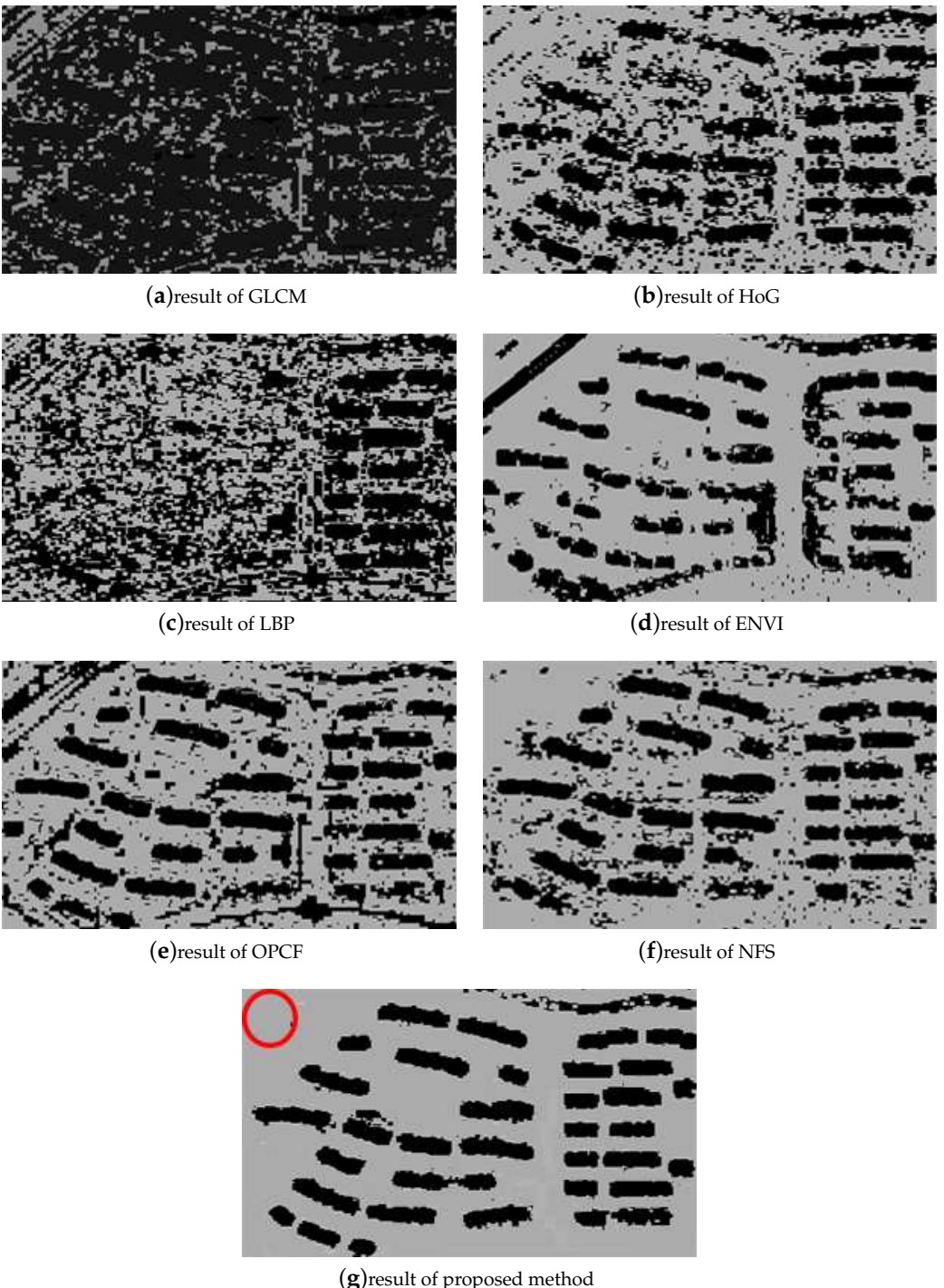

(**a**)result of GLCM

(**b**)result of HoG

(**c**)result of LBP

(**d**)result of ENVI

(**e**)result of OPCF

(**f**)result of NFS

(**g**)result of proposed method

**Figure 10.** Building extraction results of LDR 2.

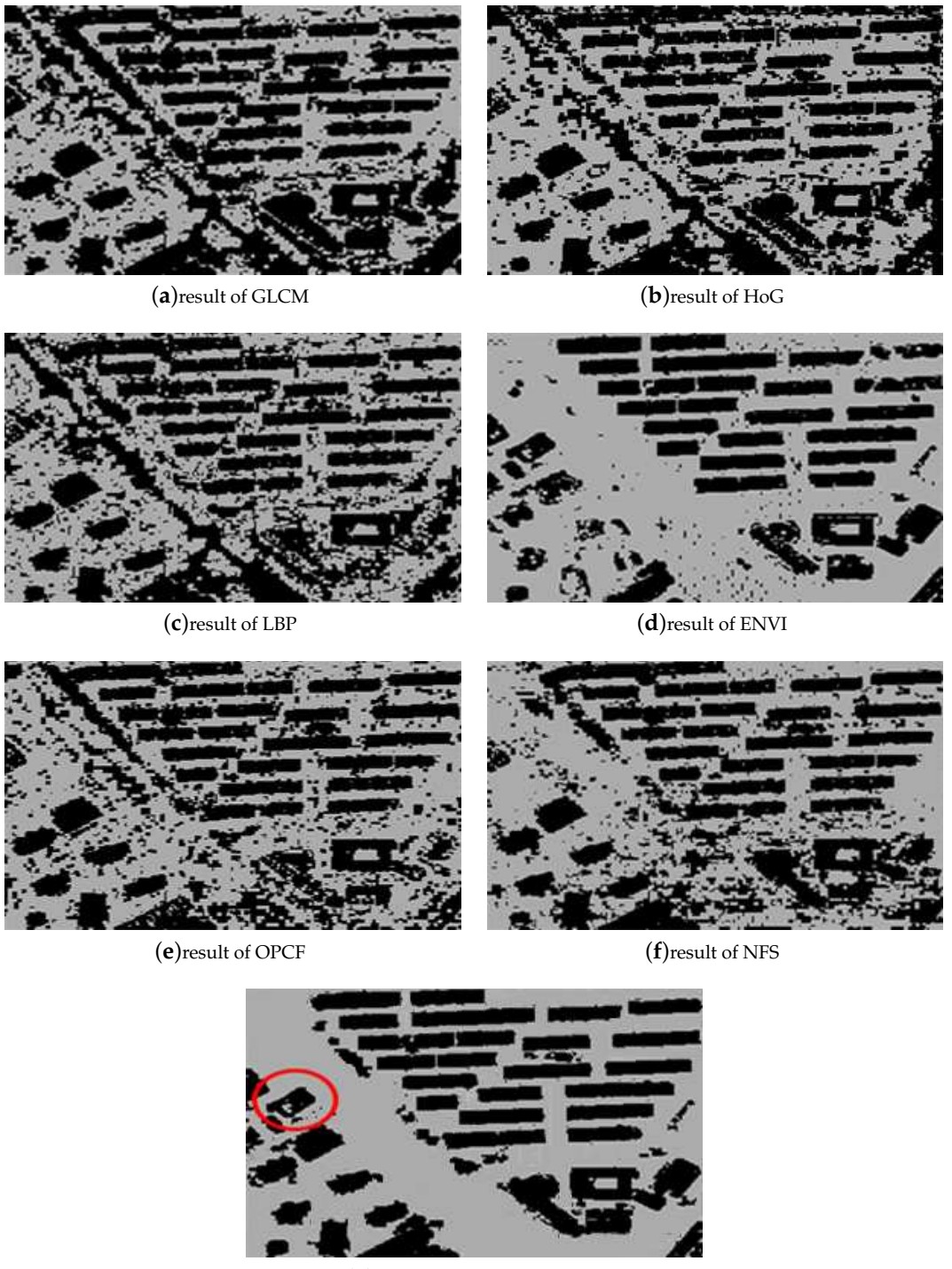

(**a**)result of GLCM

(**b**)result of HoG

(**c**)result of LBP

(**d**)result of ENVI

(**e**)result of OPCF

(**f**)result of NFS

(**g**)result of proposed method

**Figure 11.** Building extraction results of MDR.

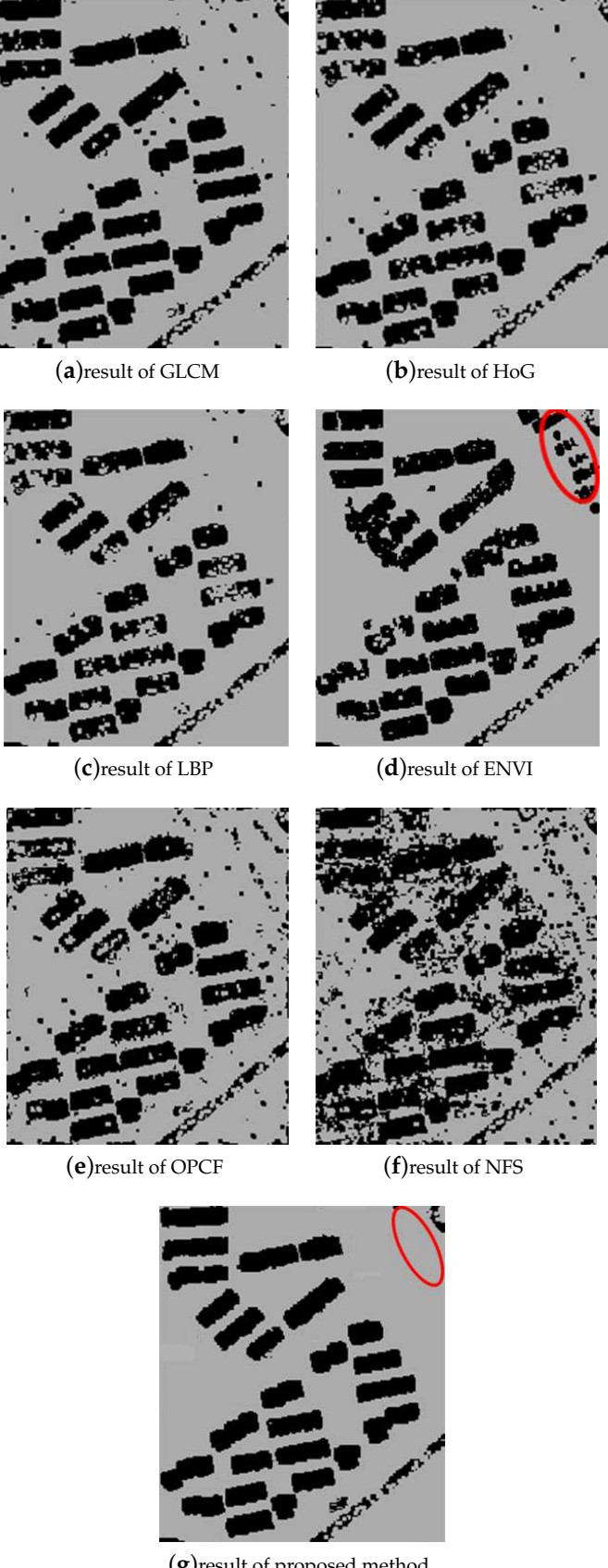

Figure 12. Building extraction results of HDR 1.

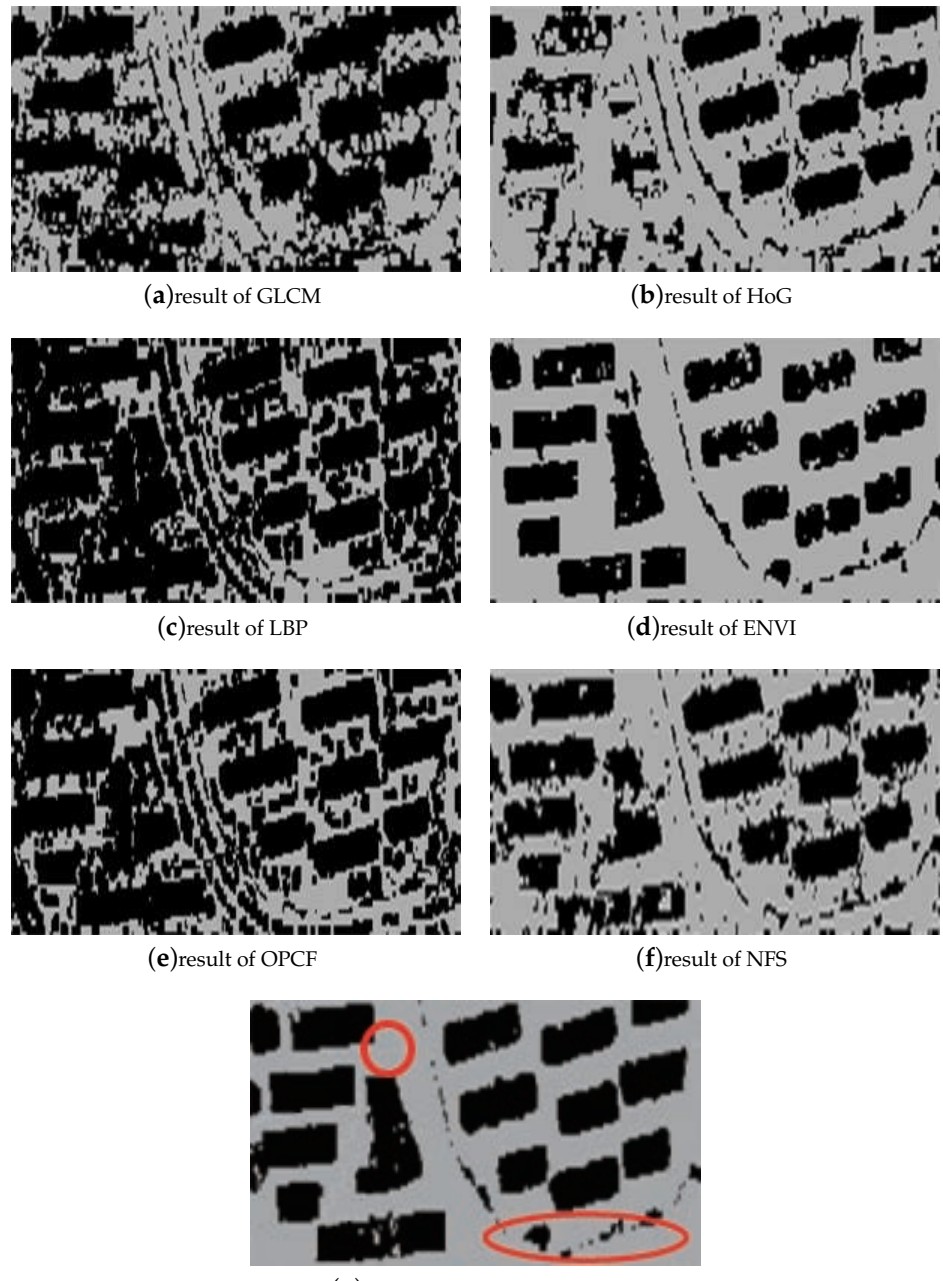

(**a**)result of GLCM

(**b**)result of HoG

(**c**)result of LBP

(**d**)result of ENVI

(**e**)result of OPCF

(**f**)result of NFS

(**g**)result of proposed method

**Figure 13.** Building extraction results of HDR 2.

As shown in Figures 9–13, the experimental results of the proposed method were superior to other texture extraction methods, such as GLCM, HoG, and LBP, as well as NFS, OPCF, and ENVI software in the five experimental areas, as it generated a lower number of errors in the building's interior area. In addition, the proposed method preserved shape better and the interior integrity of the building. HoG and NFS were incapable of extracting complete buildings in LDR1, and the proposed method was better at preserving the integrity of the large complex building on the top left corner than LBP. For LDR2, GLCM and LBP were unable to be applied for building extraction, and the proposed method produced more correct results obviously, especially in the red circle in Figure 10g, than other methods. For MDR, only ENVI software and the proposed method extracted the complete building in the red circle in Figure 11g. However, less points were extracted as building points by the proposed method in other non-building areas than ENVI software. For HDR1, all of the methods, except NFS, obtained good results in most of the test area. However, more non-building points were obviously

extracted as building points in the area of the red circle, which is shown in Figure 12d, and there were also some discrete errors in the non-building areas for other methods, while the proposed method obtained better extraction results, as Figure 12g shows. Furthermore, more buildings were extracted correctly by GLCM, LBP, and OPCF than other methods in HDR2, and for the proposed method, this was less in the red circle areas in Figure 13g.

## 5. Conclusions

This paper presented a building extraction method based on the fusion of point cloud and texture features, by calculating the feature values, elevation, and density of the point cloud and transforming the point cloud into an elevation map. The Gabor filter was used to extract texture features based on the elevation map, and the features could be assigned to the point cloud again. Then, point cloud and texture features were fused, and feature selection was done to realize more accurate and efficient building extraction. The experiments showed that the fusion of point cloud and texture features was able to obtain higher extraction accuracy than other methods. Besides, because of the large number of features, PSO was used to select a better feature combination to realize building extraction from the point cloud. Compared with the results from other building extraction methods, as well as NFS, OPCF, and ENVI software, the extraction accuracy by using the proposed method could satisfy practical applications preferably. In summary, the proposed method was proven to be efficient and valid for building extraction, with satisfactory extraction accuracy, which always exceeded 87%. It could provide a convenient and effective way to extract buildings in urban areas. On the basis of this work, future work will be performed on the optimization of the texture feature extraction method in the entire data-processing process.

**Author Contributions:** Conceptualization, X.L. and M.W.; Methodology, J.Y.; Software, J.Y. and M.W.; Validation, X.L., M.W., J.Y. and Y.L.; Formal analysis, X.L. and M.W.; Investigation, Y.L. and J.Y.; Resources, X.L. and M.W.; Data curation, X.L.; Writing—original draft preparation, X.L. and J.Y.; Writing—review and editing, X.L., M.W., J.Y. and Y.L.; Visualization, J.Y.; Supervision, X.L.; Project administration, X.L.; Funding acquisition, X.L.

**Funding:** This work was funded by the National Key Research & Development Program of China under Grant No. 41771368, the Key Laboratory for National Geographic Census and Monitoring, National Administration of Surveying, Mapping and Geoinformation under Grant No. 2018NGCM06, the Technical Research Service of Airborne LiDAR Data Acquisition and Digital Elevation Model Updating Project in Guangdong Province under Grant No. 0612-1841D0330175, and the Airborne LiDAR Data Acquisition and Digital Elevation Model Updating in Guangdong Province under Grant No. GPCGD173109FG317F.

**Conflicts of Interest:** The authors declare no conflict of interest.

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
