# Peer review of "A Building Extraction Approach Based on the Fusion of LiDAR Point Cloud and Elevation Map Texture Features"

_remotesensing, doi:10.3390/rs11141636_

Round 1

Reviewer 1 Report

The topic is building extraction from Lidar data. The method is fusion of different features from 3D point cloud and 2D elevation map. 

Overall it was well presented and tested on different dataset. However needs to have some corrections to be accepted for publication.

- English language and style of writing need to be improved.

- The abstract need to be revised and re-written in a more scientific way.

- The introduction part and literature review need to be more precisely. For example a short description about the scientific meaning of building extraction.

- The Results and Discussion section need to be revised to be clearer and fix the ambiguities.

More comments come in the attached file.

Reviewer 2 Report

Building extraction from LiDAR data is a classic in the literature, so many works concerning this topic have been published.  Then, it is difficult to find novelties about this subject, and this the case of this paper. Anyway, it has some interest that makes it feasible for publication. However, I would like to get a response to the following questions:

1)     Why don´t you use variables related to the shape of the buildings in addition to texture features.

2)     Generating an elevation map from a point cloud is not an easy task. I would like to see some comment about the quality of the DEM obtained from the point cloud.

3)     Regarding the previous item, how do you select the grid size of the elevation map?

4)     Equation (9) involves two type of objects. How do you extend this to manage all the texture or point cloud features?

5)     In lines 281-283 you says: “Besides this, the pixel values at the same corners are also different. These small differences at the edges and corners will help us to obtain multidimensional formation on buildings, which, in turn, will help us to better extract buildings” How do you implement this into your methodology?

6)     You compare the results with those obtained with ENVI software. Which algorithm used this software?

7)     The meaning of the red circles\ellipses 10-12 should be explained.

Reviewer 3 Report

Contribution:

A method for building extraction based on a fusion of point cloud features and elevation map texture features.

Major issues:

The authors have done a good job and made a useful contribution. However, I suggest they could strengthen their paper by making their contribution vs prior research clearer. This could be in the abstract or the introduction.

Minor issues:

line 36: you have "It is widely used in geodesy, geo-statistics, archaeology, geography, and the control and navigation of autonomous vehicles". Suggest adding a few references.

line 63: "extract" -> "provide"

Line 117: consider including a diagram for the Gabor steps

Line 120: "is denoised" - "was denied"

Line 177: "gird" - "grid"

Figure 1: clarify what the left image is, e.g. Elevation-colored point cloud

Equation 8, what is "rand"?

Line 230: "paper," - "paper." (comma should be period)

Line 239: sentence needs revision

Line 290: suggest you mean "extraction with the proposed method"

Line 330: add "method"

Round 2

Reviewer 2 Report

The responses to my comments are suitable. I congratulate you on your work.